# Whole genome sequencing identifies novel structural variant in a large Indian family affected with X-linked agammaglobulinemia

Abhinav Jain[1,2☯], Geeta Madathil Govindaraj[3,4☯]*, Athulya Edavazhippurath[3,5], Nabeel Faisal[3], Rahul C. Bhoyar[1], Vishu Gupta[1,2], Ramya Uppuluri[6], Shiny Padinjare Manakkad[3], Atul Kashyap[1], Anoop Kumar[1], Mohit Kumar Divakar[1,2], Mohamed Imran[1,2], Sneha Sawant[7], Aparna Dalvi[7], Krishnan Chakyar[3], Manisha Madkaikar[7], Revathi Raj[6], Sridhar Sivasubbu[1,2]*, Vinod Scaria[1,2]*

**1** CSIR-Institute of Genomics and Integrative Biology, New Delhi, Delhi, India, **2** Academy of Scientific and Innovative Research (AcSIR), Ghaziabad, Uttar Pradesh, India, **3** Department of Pediatrics, Government Medical College Kozhikode, Kozhikode, Kerala, India, **4** Department of Pediatrics, FPID Regional Diagnostic Centre, Government Medical College Kozhikode, Kozhikode, Kerala, India, **5** Multidisciplinary Research Unit, Government College Kozhikode, Kozhikode, Kerala, India, **6** Department of Pediatric Hematology, Oncology, Blood and Marrow Transplantation, Apollo Hospitals, Chennai, Tamil Nadu, India, **7** Department of Pediatric Immunology and Leukocyte Biology, ICMR-National Institute of Immunohaematology, KEM Hospital, Mumbai, Maharashtra, India

☯ These authors contributed equally to this work.
* vinods@igib.in (VS); sridhar@igib.in (SS); geetakkumar@gmail.com (GMG)

**Data Availability Statement:** The data involved in the study will be provided upon request, as it could be potentially identifiable and contains patient

## Abstract

X—linked agammaglobulinemia (XLA, OMIM #300755) is a primary immunodeficiency disorder caused by pathogenic variations in the *BTK* gene, characterized by failure of development and maturation of B lymphocytes. The estimated prevalence worldwide is 1 in 190,000 male births. Recently, genome sequencing has been widely used in difficult to diagnose and familial cases. We report a large Indian family suffering from XLA with five affected individuals. We performed complete blood count, immunoglobulin assay, and lymphocyte subset analysis for all patients and analyzed Btk expression for one patient and his mother. Whole exome sequencing (WES) for four patients, and whole genome sequencing (WGS) for two patients have been performed. Carrier screening was done for 17 family members using Multiplex Ligation-dependent Probe Amplification (MLPA) and haplotype ancestry mapping using fineSTRUCTURE was performed. All patients had hypogammaglobulinemia and low CD19+ B cells. One patient who underwent Btk estimation had low expression and his mother showed a mosaic pattern. We could not identify any single nucleotide variants or small insertion/ deletions from the WES dataset that correlates with the clinical feature of the patient. Structural variant analysis through WGS data identifies a novel large deletion of 5,296 bp at loci chrX:100,624,323–100,629,619 encompassing exons 3–5 of the *BTK* gene. Family screening revealed seven carriers for the deletion. Two patients had a successful HSCT. Haplotype mapping revealed a South Asian ancestry. WGS led to identification of the accurate genetic mutation which could help in early diagnosis leading to improved outcomes, prevention of permanent organ damage and improved quality of life, as well as enabling genetic counselling and prenatal diagnosis in the family.

sensitive information. Access to the data could be requested by mailing to Dr. Jyoti Yadav (j.yadav@igib.res.in) who is the convener of the Institutional Human Ethics Committee of CSIR-IGIB.

**Funding:** The work was supported by Council of Scientific and Industrial Research (CSIR) India through grant number MLP1801 (RareGen-CSIR India), Science & Engineering Research Board (SERB) through grant number EMR/2016/006828/HS (SERB, DST) and Foundation for Primary Immunodeficiency Diseases (FPID, USA).

**Competing interests:** The authors have declared that no competing interests exist.

## Introduction

X—linked agammaglobulinemia (XLA) is a monogenic primary immunodeficiency disorder caused by pathogenic mutations in the *BTK* (Bruton's tyrosine kinase) gene [1] with X-linked recessive inheritance. The *BTK* gene is involved in the development, maturation, and signaling of B cells [2]. The absence of plasma B cells leads to markedly reduced levels of all classes of immunoglobulins. Patients with pathogenic variants in the *BTK* gene typically manifest with recurrent infections between 3 and 18 months of age. The commonest infections are those of the respiratory tract caused by encapsulated bacteria. [3, 4]. The prevalence of XLA has been estimated to be 1 in 190,000 male births or 1 in 379,000 total live births, with 40% of cases having a positive family history [5]. As per the diagnostic criteria, the patient needs to be a male, who has hypogammaglobulinemia or agammaglobulinemia, <2 percent CD19+ B cells, and either a male family member of maternal lineage who is documented to have agammaglobulinemia and <2 percent CD19+ B cells or a confirmed (by DNA, messenger ribonucleic acid (mRNA), or protein analysis) defect in the *BTK* gene or Btk expression [6].

The *BTK* gene loci are mapped to Xq21.3-Xq22 spanning 36.7 kb on the X chromosome and consists of 19 exons and 5 functional domains. A large number of genetic variants spanning this locus have been mapped to the *BTK* loci and systematically collected and deposited in the BTKbase database http://structure.bmc.lu.se/idbase/BTKbase/ [7]. A large number of mutations reported are single nucleotide variations (73%) that include missense, nonsense, and splice site variations. Small insertions and deletions (18%) and structural variations (9.5%) account for the remaining mutations. Only 3.5% of the gross deletions reported disrupting the functionality of the *BTK* gene [8]. A large number of patients with XLA have been reported with either a deletion in only the *BTK* gene or gross deletions encompassing *BTK* along with a few neighboring genes like *TIMM8A*, *TAF7L*, *Artemis*, *IGHM*, and *DRP2* [9, 10]. The contiguous deletion in the *BTK* and *TIMM8A* gene that causes immunodeficiency with dystonia, optic neuronopathy, and sensorineural deafness is known as XLA-Mohr-Tranebjærg syndrome (XLA-MTS) [11]

We report five patients belonging to a large Indian family suffering from XLA. Whole genome sequencing identified a novel large deletion of 5,296 bp at loci chrX:100,624,323–100,629,619 spanning *BTK* exons 3–5. Screening by Multiplex Ligation-dependent Probe Amplification (MLPA) in the extended family identified seven carriers for the deletion. We surmise that the approach from next-generation sequencing to a low-cost screening method is replicable and could help to reduce the disease burden in the community.

## Materials and methods

### Patient and clinical workup

Five male children belonging to an Indian family were evaluated as a part of a programme on primary immune deficiency disorders at the Government Medical College, Kozhikode, Kerala, and the Institute of Genomics and Integrative Biology, Delhi between 2015 and 2019. The study was approved by the Institutional Ethics Committee, Government Medical College, Kozhikode and CSIR-IGIB (Ethics No. GMCKKD/RP2017/IEC/147). Written informed consent was obtained from the parents of all children who participated in the study. The index cases were P2 and P3. After elaborate pedigree analysis, clinical characteristics were recorded in a semi-structured proforma and included age at diagnosis, number of hospitalizations and PICU admissions, type of infections, complications, outcome, etc. Clinical investigations performed were complete blood count, lymphocyte subset analysis, and immunoglobulin assay.

## Btk protein expression

The cytoplasmic staining procedure was used to analyze the Btk protein expression by flow cytometry. We took 50 μL of whole blood from the patient (P2), mother, and a healthy unrelated control which was surface stained using CD3-FITC, CD14-PE for 30 min at 37˚C. An unstained tube was maintained for all three samples. At the end of incubation, these were fixed with formaldehyde for 10 minutes followed by permeabilization with Triton-X for 30 minutes at 37˚C. Subsequently, these were washed, stained with an anti-Btk monoclonal antibody for 45 minutes, washed, and analyzed by flow cytometry.

## DNA isolation and whole exome sequencing

Approximately 5 ml of blood from patients (P1, P2, P4, and P5) and their family members were drawn in an acid citrate dextrose (ACD) tube (Becton Dickinson, NJ, USA). Due to the unexpected death of P3, we could not collect his sample. Genomic DNA was isolated using the salting-out method [12] and 100 ng was used as a template to perform WES on the patient samples using the Truseq Exome library prep kit following the standard procedure as per the manufacturer's protocol (Cat no.: 20000408, Illumina Inc., SA USA). The prepared library was sequenced on the HiSeq2500 for three patients (P1, P2, and P4) and the NovaSeq6000 platform for P5 patients had paired-end read with a read length of 150 bp.

## Exome data analysis

The raw reads underwent quality control at Phred score Q30 using Trimmomatic-0.38 [13]. The trimmed reads were mapped on to human reference genome GRCh37 using Burrows-Wheeler Aligner (BWA) version 0.7.17 [14]. The mapped reads were further sorted and duplicate reads were removed using SAMtools [15] and Picard respectively https://broadinstitute.github.io/picard/. The variant calling was done using the HaplotypeCaller of Genome Analysis ToolKit (GATK version 3.8.0) best practices [16]. The variants were further systematically annotated using ANNOVAR (2018-04-16 00:47:49) [17] that comprises of multiple datasets i.e. refGene, dbsnp (avsnp150), dbnsfp35a, and clinvar 20190305. It also annotates allele frequency from the global population datasets i.e. 1000 Genome project (1000g2015_all), Genome Aggregation Database (gnomAD V2.1.1), and Esp6500. The protein altered variants (missense, stop gain, stop loss, frameshift, non-frameshift, splicing, small insertion, and small deletion) were prioritized. Further, variants whose minor allele frequency was greater than 5% in the global population were filtered out. A gene filter was applied which comprised 454 genes known to be implicated in PID and were recently catalogued by the International Union of Immunological Societies (IUIS) expert committee [18]. Finally, based on the phenotype, we correlated the variant's clinical significance.

## Whole genome sequencing and analysis

The whole genome sequencing was performed using 100 ng of genomic DNA from two patients (P1 and P4). WGS was performed with 150 bp paired-end reads that were generated using Truseq PCR free library kit as per manufacturer's instructions (Cat no.: FC-121-9006DOC Illumina Inc. SA USA) on Illumina NovaSeq 6000 platform (San Diego, CA, USA) using sequencing by synthesis chemistry. The raw reads to variant annotations were performed similarly to the whole exome analysis. We merged both the individuals' variants using the GATK (version 3.8.0) tool called CombineVariants. We filtered out the variants with MAF>0.05 in the global population. Further, we adopted an overlap based strategy [19] where we prioritized the common homozygous variants between P1 and P4. Further, variants were

filtered based on the in-silico tool CADD (Combined Annotation-Dependent Depletion) score > 15 [20]. For the remaining variants, we manually correlated phenotypic characteristics of patients with filtered variants.

We have also performed structural variant (SV) analysis on the whole genome sequenced aligned reads of both the patients (P1 and P4). We used LUMPY (version 0.2.13) for SV calling [21] that were further genotyped using SVtyper [22]. The SVs were further prioritized using an overlap-based strategy where we prioritized a common homozygous variant in the 454 PID genes [19]. To validate the SV result, we adopted the manual coverage-based analysis on WGS paired-end reads that were aligned on the reference genome hg19/GRCh37 using Integrated Genome Viewer (IGV) [23].

## Multiplex-ligation dependent probe amplification (MLPA) assay

To identify and validate the gross deletion in extended family members, the MLPA based approach was adopted. We have performed the test on 17 members of the family with their consent and institutional ethical approval (Ethics No. GMCKKD/RP2017/IEC/147). Genomic DNA was isolated using a standard salting-out method [12] and 100 ng of the genomic DNA was used. MLPA was performed as per manufacturer's instructions (MRC-Holland, Amsterdam, The Netherlands) using SALSA MLPA EK1 reagent kit-FAM (EK1-FAM, MRC Holland, Netherlands) along with the SALSA MLPA Probemix P210 BTK (P210-050R, MRC Holland, Netherlands). Capillary electrophoresis of the amplicons was performed on ABI 3130 genetic analyser (Applied Biosystems™, California, USA). MLPA data analysis was performed using Coffalyser.net software (MRC Holland, Netherlands).

## Ancestry haplotype mapping using chromosomal painting

Since it is a novel deletion and number of primary immunodeficient variants have been previously shown to have founder effects and specific ancestries [24, 25], we explored the haplotype similarity pattern of our patients with that of the global population. We used the haplotype prediction tool fineSTRUCTURE [26] (version 2.1.3) on P1 and P4 whole genome sequenced variants, whose chrX variants have been extracted and merged it with chrX variants of 1000 Genome Project as a reference and 44 whole genome data of the Qatar population. The 1000 Genome project consists of 2504 individuals from five major populations African (AFR), American (AMR), East Asian (EAS), European (EUR), and South Asian (SOU) [27]. We pruned the merged VCF by applying a variant filter of allele frequency greater than 1% and allele number filter of 1,000 to get the maximum genotype rate using a bespoke bash script. We phased the merged VCF using the SHAPEIT v2.r900 tool [28]. We ran fineSTRUCTURE pipeline which involves four steps that include painting of chromosome depending on the population haplotype with the individual i.e. chromopainter then combining all painted data and assigning a population to a block which involves chromocombine and fineSTRUCTURE respectively, and finally tree building based on maximum posterior population inference. The region 50KB, 500KB, 5MB, and 50MB upstream and downstream on both sides of the chromosomal locus chrX:100624323–100629619 has been plotted (hg19/GRCh37) using R scripts provided by fineSTRUCTURE.

## Results

### Clinical details

The five male children presented to the Government Medical College, Kozhikode, Kerala with recurrent respiratory infections, diarrhea, pyoderma, and pyogenic meningitis with onset

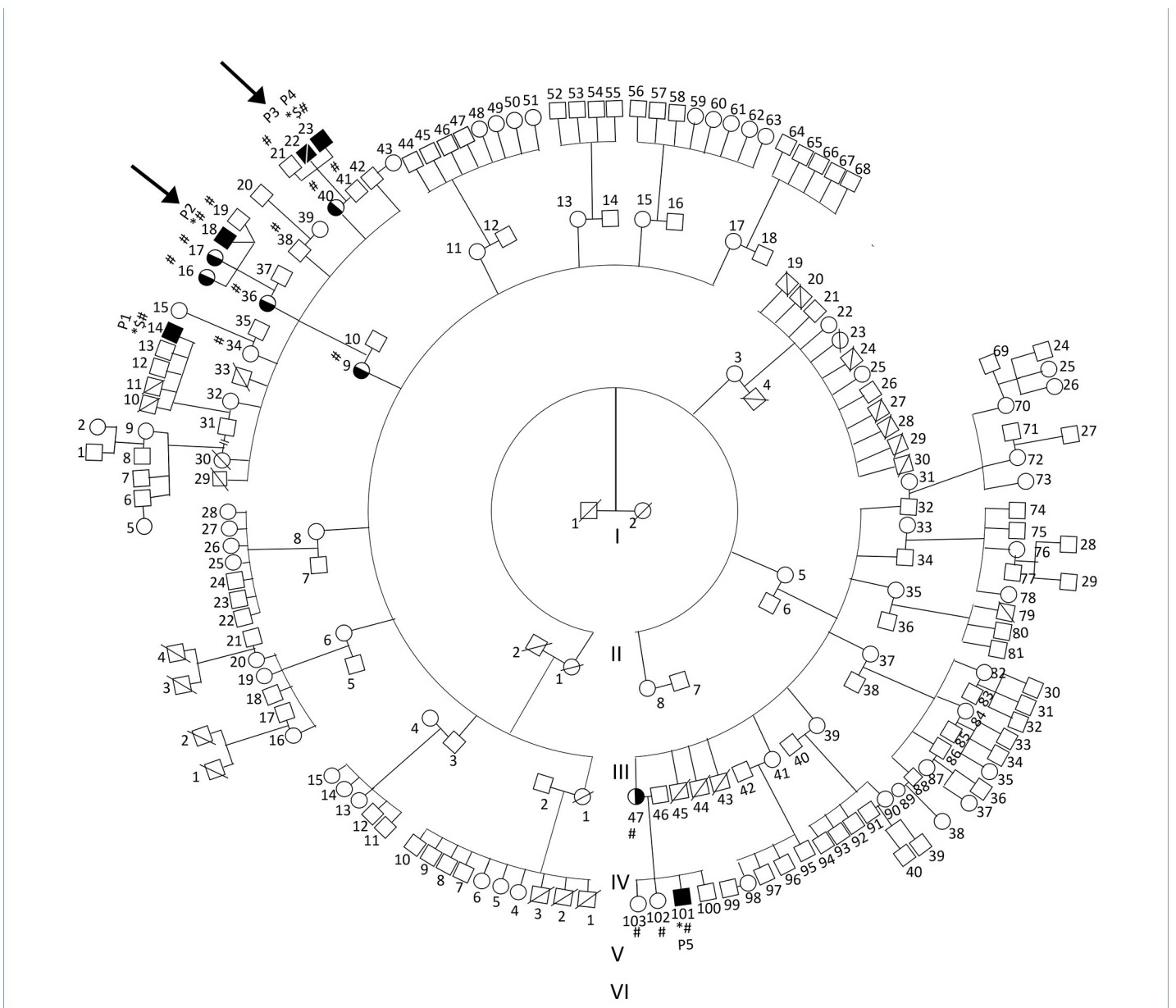

**Fig 1. Extended family pedigree of the patients affected with X-linked agammaglobulinemia.** Arrow represents two index cases, *—Individuals who underwent whole exome sequencing, $- Individuals who underwent whole genome sequencing, #- Individuals who underwent MLPA.

between 5 months and 12 months of age. They belonged to a large extended family as shown in **Fig 1**. Clinical features have been tabulated in **Table 1** and detailed in the **S1 File**.

All the children had absent or atrophic tonsils, hypogammaglobulinemia, severely reduced or absent CD19 counts, and were started on intravenous immunoglobulin (IVIG) infusions. Two children who presented late, and were not on regular prophylaxis with IVIG developed bronchiectasis. Other complications included stunted growth, delayed puberty, and arthritis. One child died soon after diagnosis due to meningitis at 20 months of age. Analysis of the pedigree chart revealed that 22 male children had died in the early years of life as shown in **Fig 1**.

**Table 1. Clinical and Immunological features of patients with XLA in the present cohort.**

| Clinical and immunological characteristics | P1 | P2 | P3 | P4 | P5 |
|---|---|---|---|---|---|
| Age at onset | 1 year | 5 months | 7 months | 5 months | 6 months |
| Age at diagnosis | 7 years | 1 year 7 months | 1 year 6 months | 6 months | 8 years |
| Type of infections | Pneumonia, Pyoderma, Pyogenic meningitis | Pneumonia, Acute suppurative otitis media, Diarrhea, Oral thrush | Pneumonia Diarrhea Pyoderma | Diarrhea | Pneumonia, Pyogenic meningitis Diarrhea, pyoderma |
| Absent / atrophic tonsils | yes | yes | yes | yes | yes |
| Sibling death | yes | no | yes | yes | no |
| No. of hospitalizations before diagnosis | 2 | >15 | 1 | 2 | >10 |
| No. of PICU admissions | none | 2 | 1 | none | 1 |
| Immunoglobulin assay | IgG low, IgA low, IgM low | IgG low, IgA low, IgM normal | IgG low, IgA normal, IgM normal | IgG low, Ig A low, IgM normal | IgG low, IgA low, IgM normal |
| CD 19 count at diagnosis (N) In cells/mm3 | 0 (91–610) | 0 (430–3300) | 8 (430–3300) | 10 (430–3300) | 10 (91–610) |
| CD 3 count at diagnosis (N) in cells/mm3 | 6458 (570–2400) | 5475 (1460–5440) | 6400 (1460–5440) | 4169 (1900–5900) | 9553 (570–2400) |
| CD 56 count at diagnosis (N) in cells/mm3 | 355 (78–470) | 7.0175 (80–340) | 160 (80–340) | 661 (160–950) | 528 (78–470) |
| Onset to IVIG start | Not on regular IVIG | 1 year, 2 months | 11 months | 1 month | Not on regular IVIG |
| Complications | Bronchiectasis | None | None | None | Bronchiectasis, Arthritis |
| Outcome | Stunted growth, | Stunted growth, Cured–HSCT from matched sibling donor (twin) at 6.5 years | Stunted growth Died | Cured-HSCT from matched sibling donor at 2 years | Stunted growth and delayed puberty |

## Btk expression estimation using flow cytometry

By using this flow cytometric approach, patient P2 has low Btk expression observed on monocytes (37%) compared to 87% in the control, the normal range being 90 +/- 5%. His mother showed a mosaic pattern suggestive of being a carrier for XLA.

## Whole exome sequencing analysis

We performed whole exome sequencing for four patients (P1, P2, P4, and P5) with 99.5% alignment on the human reference genome for each patient and average coverage of 90.8X. A total of 467,638 variants were called for each patient. Variants were annotated using a tool ANNOVAR [17] on average, 13,074 protein-altering variants consisting of splicing, exonic splicing, and exonic variants except synonymous SNVs were prioritized. These variants were further prioritized for minor allele frequency (MAF) < 5% in the global population which reduces the average number of variants to 2,383. On average 12 genetic variants were mapped to 454 primary immunodeficiency genes, for further downstream analysis. We could not find any variant which could correlate with the clinical features. The whole exome sequencing data with variant filtering has been tabulated in **S1 Table**.

## Whole genome sequencing analysis

Since we could not find any causal variant using whole exome sequencing, we performed whole genome sequencing of P1 and P4 with a mapping percentage of 99.58% and 99.49% and coverage of 21X and 86X respectively. Variant calling was done by GATK (version 3.8.0)

**Table 2. Whole genome sequencing structural variant analysis for P1 and P4.**

| Data | P1 | P4 |
|---|---|---|
| Total Structural Variants | 8,925 | 29,638 |
| Common homozygous Variants | 768 | |
| PID genetic variants (454 gene) | 8 | |
| Phenotype associated variant | 1 | |
| Structural Variant | chrX:100,624,323–100,629,619 (5296 bp deletion) | |

HaplotypeCaller and resulted in 4,905,687 and 4,904,616 variants for P1 and P4 respectively. The variant files of both the patients P1 and P4 were merged and annotated using a tool ANNOVAR [17] which led to a total of 6.5 million variants. These variants were prioritized for minor allele frequency (MAF) < 5% in the global population dataset, that reduces the variant number to 890,387. Further, we adopted an overlap-based strategy [19] and prioritized 27,236 variants. On applying in-silico computational tool i.e. CADD score > 15, we prioritized 4 variants. On manual correlation with the filtered variants, we could not correlate any of the prioritized variants with the clinical characteristics of the patients. The whole genome sequencing data with variant filtering has been tabulated in **S2 Table.**

We have also performed structural variant analysis using LUMPY for calling SV. We got a total of 8,925 and 29,638 SVs in patient P1 and P4 respectively. Prioritizing common and homozygous SVs in both the patients, led to a total of 768 SVs. Further we applied the PID gene filter that drops the number to 8 SVs. Finally, on clinical correlation with SVs, we prioritized a novel large deletion of 5,296 bp on chrX at loci ranging from 100,624,323 to 100,629,618. The variant filtering at each step has been tabulated in **Table 2.** Further visualizing on the human reference genome hg19/GRCh37 using IGV, we found that there were no reads present on chrX at loci ranging from 100,624,323 to 100,629,618 i.e. 5296 bases (~5 Kb). This led to the identification of a large deletion spanning exon 3 to exon 5 of the *BTK* gene for both P1 and P4 as shown in **Fig 2**. However, the region flanking the deleted loci showed adequate mapped reads. The deleted region in the patient has also been properly covered in the control sample. This prompted us to visualize the whole exome data for P2 and P5 on IGV and as a result, we found the same *BTK* gene deletion encompassing exon 3–5 as shown in **S1 Fig**. On intersecting patients deleted loci chrX:100624323–100629619 with the 1000 Genome project SVs, gnomAD SVs, and IndiGen SVs (in-house) database, we could not find any structural variant that falls in the exon 3–5 of *the BTK* gene in any of the databases.

## Hematopoietic stem cell transplantation (HSCT)

After the family was counseled regarding the feasibility of HSCT as a curative option, two patients P2 and P4 underwent HSCT from a matched sibling donor. They required myeloablative conditioning to prevent graft rejection, and a treosulfan-based reduced toxicity protocol to ensure adequate myeloablation. Graft versus host disease prophylaxis consisted of short-course methotrexate and tacrolimus. Both of them are now disease-free.

## Family screening using the multiplex ligation-dependent probe amplification (MLPA) assay

We performed whole genome sequencing on two samples P1 and P4 and found a large hemizygous deletion encompassing 3–5 exons of the *BTK* gene. To confirm whether the mutation is *de novo* or inherited, MLPA assay-based approach was adopted. The assay was first validated using the P1 and P4 samples as positive controls and two reference samples from healthy

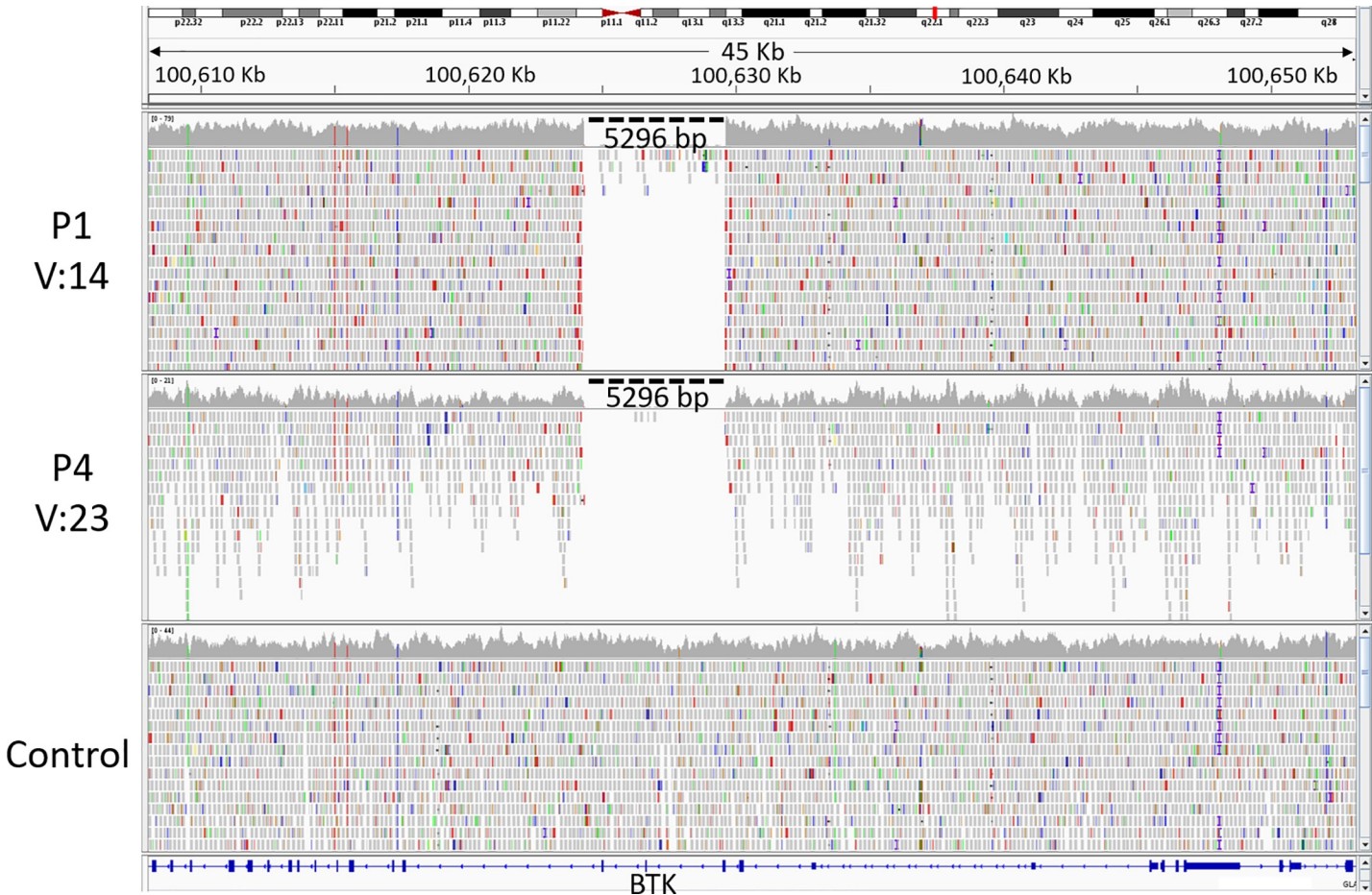

**Fig 2. Identification of putative large deletion in whole genome data in P1 and P4.** The figure represents the visualization of sequencing reads aligned on the coding region of the *BTK* gene.

individuals. The positive control samples P1 and P4 were found to have the probe ratio of zero for the exons 3–5 of *BTK* gene i.e. there is the absence of any copy for this region as shown in **Fig 3** and therefore, the deletion was detected, which corroborated with the whole genome sequencing results shown in **Fig 2**.

The standardized and validated MLPA assay was then used to screen additional family members of P1 and P4 for the deletion in the *BTK* gene. The probe ratios for the mothers of both P1 and P4 (IV-32 and IV-40) corresponded to ~0.5 as shown in **S2 Fig**, therefore, they were heterozygous for the same deletion. This shows that the deletion has been inherited by both P1 and P4 from their mothers. The brother (V-21) and father (IV-41) of P4 have also been tested and were found to have a normal copy of the gene. The other two patients, P2 and P5 were tested using the MLPA based assay and were found to harbor the same deletion in hemizygous form as shown in **Fig 3**. Following this, the families of P2 and P5 were tested. The maternal grandmother (III-9), mother (IV-36), and the two sisters (V-16 and V-17) of P2 were found to be heterozygous for the deletion, and one brother (V-19) was found to be carrying a normal copy of the gene **S2 Fig**. The mother (III-47) of P5 was found to be heterozygous for the deletion and her two sisters (IV-102 and IV-103) were found to have inherited both the normal copies of the gene. Additionally, two other members (V-34 and V-38), one male and one female, of the extended family were tested for the deletion and were found to have normal

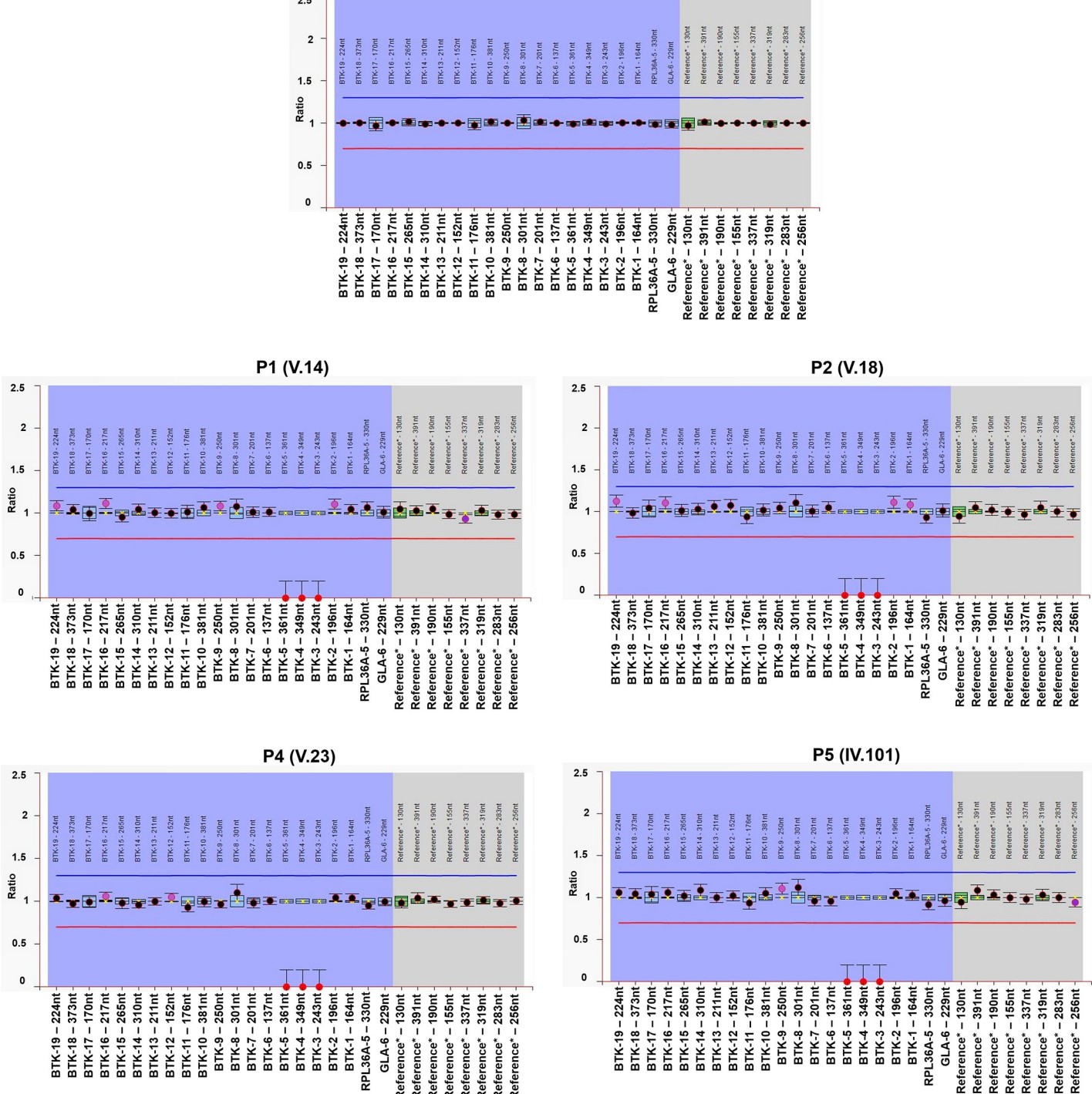

**Fig 3. Multiplex ligation dependent amplification (MLPA) assay for detection of deletion encompassing 3–5 exons of the *BTK* gene: Representation of ratio charts generated using the Coffalyser.net software.** (A) Reference sample depicts the normal copies of the BTK gene and other samples (B) Sample P1 (C) Sample P2 (D) Sample P4 (E) Sample P5 are all males and depict probe ratio of zero for 3–5 exons of the BTK gene which lies on X-chromosome, i.e. they have hemizygous deletion. For all the ratio charts, the longitudinal axis (X-axis) represents the final ratios after inter and intra normalisations of the probe ratios. Horizontal axis (Y-axis) represents the probe names alongwith the length in the order (Both axis title have been manually enlarged for clarity). The blue and red horizontal lines depict the arbitrary borders of ratio 1.3 and 0.7 respectively. The black and red dots denote the final ratio obtained for each of the probes and the vertical bars represent the 95% confidence range for each probe. The Roman and numeric numbers on top of each ratio chart represent the individual marked as per the pedigree in Fig 1.

copies of the gene. Using MLPA-based assay, it has been found that the large deletion is inherited in multiple members of the family. Out of 17 members tested, we found four members of the family were hemizygous and seven were carriers for the deletion. Upon analysis of the pedigree in **Fig 1**, we can predict more than half a dozen of the family members could be carriers of the deletion. The MLPA probe ratio for each individual at 19 *BTK* exons test probes as well as for reference probes has been tabulated in **S3 Table.**

## Chromosomal painting analysis

Since this variation is absent in all the control populations, it excites us to know the haplotype ancestry around the deleted region. In order to predict haplotype ancestry of the region flanking the deletion on chromosome X, we merged chrX variants of whole genome sequenced P1 and P4 individually with 2,504 individuals of five major populations from 1000 Genome Project and 44 whole genome sequenced Qatari individuals which comprise 3,821,263 variants. We pruned the variants whose allele frequency is less than 0.01 and allele number less than 1,000 that filtered the variant number to 185,258, and 186,417 for P1 and P4 respectively. After phasing and chromosomal painting using fineSTRUCTURE, we found that both the samples had South Asian ancestry. The painted chromosomal region of both the patients P1 and P4, 5MB upstream and downstream of the deleted locus has been well represented in **Fig 4**. For more fine visualization we have painted the chromosomal region at 50KB, 500KB, and 50MB upstream and downstream to the deleted locus in **S3 Fig**.

## Discussion

XLA was the first primary immunodeficiency disorder discovered by a pediatrician Ogden Carr Bruton in 1952 [29]. The onset of symptoms is usually before one year of age. Infections do not occur in early infancy due to the protective effect of maternally derived IgG [5]. Apart

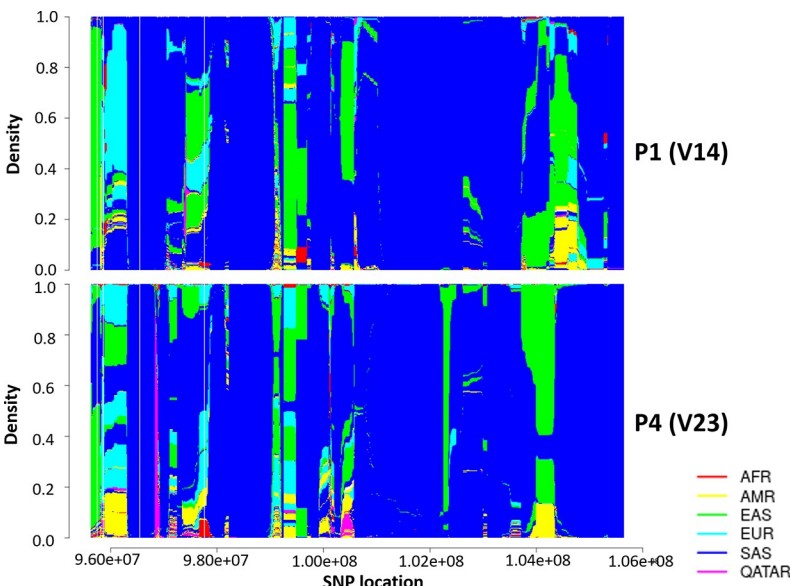

**Fig 4. Haplotype ancestry prediction with the global population using fineSTRUCTURE.** South Asian ancestry predicted the haplotype for locus 5MB upstream and 5MB downstream to loci chrX:100,624,323–100,629,619 (hg19/GRCh37) of two affected patients (V14 and V23) with 2504 individuals of five major populations (AFR-African, AMR-American, EAS- East Asian, EUR-European, and SAS-South Asian) of 1000 Genome Project and 44 individuals of Qatari ancestry.

from an increased frequency and severity of respiratory and gastrointestinal infections, there are other rare manifestations described including arthritis, which was a feature in P5, and significantly impinges on the quality of life [30]. Autoimmunity and autoinflammation can occur in XLA, and arthritis could be either due to a dysfunctional immune system or due to infection with mycoplasma, which was not the case in our patient P5 [30]. The challenge lies in identifying the index case [31]. Although panhypogammaglobulinemia is the rule, there are reports of normal immunoglobulin levels or selective immunoglobulin deficiency [32, 33]. Antibody responses to isohemagglutinins and vaccine antigens were not performed since B cell numbers were negligible, and it was possible to look for a *BTK* variant.

Identifying the variant is critical for carrier detection and genetic counseling, without which the disease frequency in the community cannot be reduced. We have performed WES in all the four patients but unfortunately were unable to find the causal variant. This could be due to the limitation of WES in providing adequate coverage for all the genes [34] inability in calling structural variants [35], missing non-coding variants [36] as well as high-quality coding SNVs, [37]. The WGS has paved the way for the diagnosis of such difficult-to-diagnose disorders. In the present study, we identified a novel large deletion of 5,296 bp encompassing exons 3–5 of the *BTK* gene in four patients. A very recent analysis was done by Thaventhiran and group, where they performed WGS analysis on 1,318 PID patients and identified eight structural variants which could have been missed by WES [38]. WGS also inferred *denovo* 3Mb deletion breakpoints in two probands affected by DiGeorge syndrome [39].

XLA is caused by pathogenic variants in the *BTK* gene, which is a cytosolic tyrosine kinase protein composed of five domains i.e. pleckstrin homology (PH), Tec homology (TH), Src homology domain (SH2), SH3, and kinase (Catalytic) domains [40]. In our patients, the 5,296 bp deletion spans exons 3–5 of *BTK* gene. Exon 3 is an integral part of the PH domain, which regulates the binding of molecules like inositol 1,3,4,5-tetrakisphosphate ($IP_4$), inositol 1,3,4,5,6-pentakisphosphate ($IP_5$), and inositol 1,2,3,4,5,6-hexakisphosphate ($IP_6$) which in turn activates *BTK* gene function [41]. There are multiple mutations reported in exon 3 which affect the functionality of the PH domain [42] and cause XLA. Mutations in exon 4 which encode for the C-terminal immunoglobulin domain [43] and mutations in exon 5 also have been reported in a patient with decrease in Btk expression and CD19+ B cell number [44].

Early diagnosis is imperative to avoid morbidity, mortality and vaccination with OPV, since these children are at risk for paralytic poliomyelitis. [45]. In the family studied, 22 male children had died due to infections as shown in **Fig 1**. These individuals might have been harbouring the same hemizygous deletion and early diagnosis could have been life-saving. Using a low-cost MLPA-based assay for screening additional family members, we have found seven female carriers for the *BTK* exon 3–5 deletion. Unfortunately, we were able to test only 17 out of 159 individuals. But by analyzing the pedigree, we can predict that more than half a dozen family members could be carriers. Carrier detection in families affected by Mendelian disorders will enable genetic counseling and antenatal diagnosis, ultimately resulting in a reduced disease burden. [46, 47]. MLPA has been used to screen for multiple diseases as well as prenatal screening, due to its swift, highly sensitive and cost-effective approach [48, 49]. Both children in whom the diagnosis was delayed and who were not on regular IVIG prophylaxis (P1 and P5) developed permanent lung damage.

There are reports of successful HSCT for XLA, but it has not been used extensively since immunoglobulin replacement is widely available. However, when the cost and availability of lifelong prophylaxis is a limiting factor, HSCT has been chosen by parents instead of the option of lifelong immunoglobulin replacement [50, 51]. This was the case in P2 and P4 patients.

While the mean age at onset was 7 months, the mean age at diagnosis was 44.6 months, a delay of 37.6 months, resulting in multiple hospital admissions for treatment of infections.

Clinician and patient/parent education would help to ensure early diagnosis, enhance compliance with treatment and prevent poor outcomes as occurred in P1 and P5. Transitioning to adult services is also a challenge [52]. Accurate genetic workup and counseling of the extended family will reduce the burden of care. [53].

The WGS and the low-cost MLPA assay are replicable in the society for reducing the disease burden of affected families and the community at large. The effectiveness of this approach hinges on the availability and accessibility of a system for genetic counseling that the community would accept.

## Supporting information

**S1 Fig. Visualization of putative large deletion encompassing *BTK* gene exon 3–5 of whole exome data of patients P1, P2, P4, P5, and control sample.**
(TIF)

**S2 Fig. MLPA for detection of deletion encompassing 3–5 exons of the *BTK* gene in the additional family members of the proband.** Representation of ratio charts generated using the Coffalyser.net software. Longitudinal axis represents the final ratios after inter and intra normalisations of the probe ratios. Horizontal axis represents the probe names along with the length (The axis titles have been manually enlarged for clarity). The blue and red horizontal lines depict the arbitrary borders of ratio 1.3 and 0.7 respectively. The black and red dots denote the final ratio obtained for each of the probes and the vertical bars represent the 95% confidence range for each probe. The test probes are of *BTK* gene and the rest are the reference probes. The Roman and numeric numbers on top of each ratio chart represent the individual marked as per the pedigree in **Fig 1**.
(TIF)

**S3 Fig. South Asian ancestry predicted for locus.** A) 50MB upstream and 5KB downstream, B) 500KB upstream and 500KB downstream, and C) 50KB upstream and 50KB downstream to loci chrX:100,624,323–100,629,619 (hg19/GRCh37) of two affected first cousins (V14 and V23) with 2504 individuals of five major populations (AFR-African, AMR- American, EAS-East Asian, EUR-European, and SAS-South Asian) of 1000 Genome Project and 44 individuals of Qatar ancestry.
(TIF)

**S1 Table. Whole exome sequencing data summary for each patient.**
(PDF)

**S2 Table. Whole genome sequencing data summary for P1 and P4.**
(PDF)

**S3 Table. MLPA copy number ratios for each of the reference and test probes of the proband and all the extended family members tested.** The roman and numeric numbers on top of each ratio chart represent the individual marked as per the pedigree in **Fig 1**.
(PDF)

**S1 File. Detailed clinical feature of XLA large deletion family.**
(DOCX)

## Acknowledgments

We thank Disha Sharma, Bani Jolly, Mukta Poojary, and Afra Shamnath, for suggestions which enriched the manuscript.

## Author Contributions

**Conceptualization:** Abhinav Jain, Geeta Madathil Govindaraj, Sridhar Sivasubbu, Vinod Scaria.

**Data curation:** Abhinav Jain, Geeta Madathil Govindaraj, Athulya Edavazhippurath.

**Formal analysis:** Abhinav Jain, Geeta Madathil Govindaraj, Athulya Edavazhippurath, Nabeel Faisal, Rahul C. Bhoyar, Vishu Gupta, Ramya Uppuluri, Shiny Padinjare Manakkad, Atul Kashyap, Anoop Kumar, Mohit Kumar Divakar, Mohamed Imran, Sneha Sawant, Aparna Dalvi, Krishnan Chakyar, Manisha Madkaikar, Revathi Raj.

**Funding acquisition:** Geeta Madathil Govindaraj, Sridhar Sivasubbu, Vinod Scaria.

**Investigation:** Abhinav Jain, Geeta Madathil Govindaraj, Nabeel Faisal, Anoop Kumar, Vinod Scaria.

**Methodology:** Abhinav Jain, Geeta Madathil Govindaraj, Athulya Edavazhippurath, Rahul C. Bhoyar, Vishu Gupta, Ramya Uppuluri, Shiny Padinjare Manakkad, Atul Kashyap, Mohit Kumar Divakar, Mohamed Imran, Revathi Raj.

**Project administration:** Geeta Madathil Govindaraj, Sridhar Sivasubbu, Vinod Scaria.

**Resources:** Geeta Madathil Govindaraj, Sridhar Sivasubbu.

**Software:** Abhinav Jain.

**Supervision:** Geeta Madathil Govindaraj, Sridhar Sivasubbu, Vinod Scaria.

**Validation:** Abhinav Jain, Athulya Edavazhippurath.

**Visualization:** Abhinav Jain, Athulya Edavazhippurath.

**Writing – original draft:** Abhinav Jain, Geeta Madathil Govindaraj.

**Writing – review & editing:** Abhinav Jain, Geeta Madathil Govindaraj, Sridhar Sivasubbu, Vinod Scaria.

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
