## [Decision Letter · Decision Letter 0]

16 Mar 2021

PONE-D-20-38741

Whole genome sequencing identifies novel structural variant in a large Indian family affected with X-linked agammaglobulinemia

PLOS ONE

Dear Dr. Scaria,

Thank you for submitting your manuscript to PLOS ONE. After careful consideration, we feel that it has merit but does not fully meet PLOS ONE’s publication criteria as it currently stands. Therefore, we invite you to submit a revised version of the manuscript that addresses the points raised during the review process.

Based on internal evaluation and external review this manuscript was found to be interesting. However the reviewers raised points that needs to be addressed carefully, especially the claim that WES did not give any casual variants was not taken well by one of the reviewers and suggested to look for CNVs before concluding. Please try to address all those points by performing the analysis or write rebuttal putting forward your arguments.

We look forward to receiving your revised manuscript.

Kind regards,

Obul Reddy Bandapalli, MSc, PhD

Academic Editor

PLOS ONE

Journal Requirements:

2. Thank you for including your ethics statement:  "The study was approved by the Institutional Ethics Committee (Ethics No. GMCKKD/RP2017/IEC/147).".   

3. Please provide additional details regarding participant consent.

In the ethics statement in the Methods and online submission information, please ensure that you have specified what type you obtained (for instance, written or verbal, and if verbal, how it was documented and witnessed).

If your study included minors, state whether you obtained consent from parents or guardians.

If the need for consent was waived by the ethics committee, please include this information.

Reviewers' comments:

Reviewer's Responses to Questions

**Comments to the Author**

1. Is the manuscript technically sound, and do the data support the conclusions?

Reviewer #1: Partly

Reviewer #2: Yes

2. Has the statistical analysis been performed appropriately and rigorously? 

Reviewer #1: Yes

Reviewer #2: N/A

3. Have the authors made all data underlying the findings in their manuscript fully available?

Reviewer #1: Yes

Reviewer #2: Yes

4. Is the manuscript presented in an intelligible fashion and written in standard English?

Reviewer #1: Yes

Reviewer #2: Yes

5. Review Comments to the Author

Reviewer #1: The authors report the detection of a large deletion in the BTK gene in a large family with X-linked agammaglobulinemia. More than 500 mutations in this gene have previously been identified to cause the disease, so the paper adds one more mutation to the list. The paper is clearly written and the data analysis is well described. The authors have attempted to show that WGS is needed for identifying the pathogenic variant, however, the data presented does not support this claim (see below).

My main criticism is that the authors performed whole-exome sequencing on four affected individuals but say that did not find "any causal variant using whole-exome sequencing". However, this is not well justified since the authors only looked at small variants identified from exome data. Given the strong association of mutations in BTK with the phenotype, the authors could easily have looked for copy number variants specifically in the BTK gene (the deletion would easily be seen as zero read depth on the X chromosome in males). There are many tools for detection of CNVs from exome data that could also have been used. Simply visualizing read depth using the IGV viewer can reveal deletions that overlap coding regions. Therefore, the claim that WGS was needed to detect the deletion in the BTK gene is not supported. WGS is definitely beneficial for identifying the precise breakpoints but exome sequencing is still 3-4 times cheaper than WGS. I would recommend that the authors report the analysis of read depth from exome data.

The rationale for performing chromosome painting analysis of the region around the deletion is not clear to me. The large deletion identified in this family cannot be expected to be seen in controls since it would cause a severe disease in males. Therefore, it cannot be expected to have a specific population origin. The individuals studied in this paper are from India and it is not surprising that they have South Asian ancestry. This should be true for their entire genome, rather than just the region around the deletion.

Reviewer #2: A. Is the manuscript technically sound, and the data support the conclusions?

The manuscript by Jain et al. describes about the identification of large deletion (5,296 bp) encompassing exons 3-5 of the BTK gene of the patients with X - linked agammaglobulinemia (XLA) using the whole-genome sequencing (WGS). The authors have performed whole-exome sequencing (WES) and WGS of the four and two patients with XLA, respectively. Also, the authors have performed several assays including multiplex-ligation dependent probe amplification (MLPA) assay. The manuscript can be accepted after the following minor concerns have been addressed.

In abstract section, the authors must rephrase the following sentence as they concluded (in the results section) that they could not find any variant which could correlate with the clinical features: “Whole genome sequencing led to identification of the accurate genetic mutation which could help in early diagnosis leading to improved outcomes, prevention of permanent organ damage and improved quality of life, as well as enabling prenatal diagnosis”.

In the methodology section the authors can use multiple algorithms for the identification of alterations including copy number alterations using WES and WGS data of the patients with XLA.

B. Has the statistical analysis been performed appropriately and rigorously?

The statistical analysis has not been performed by the authors.

C. Have the authors made all data underlying the findings in their manuscript fully available?

The authors have provided analyzed data in the figures/tables and supplementary information in the manuscript wherever necessary. The raw data can be accessed through the convener of the Institutional Human Ethics Committee of the institute.

D. Is the manuscript presented in an intelligible fashion and written in standard English?

Overall manuscript is okay written, however the authors can rephrase the sentences in methodology section for example, “small indels, small insertion, and small deletion”.

6. PLOS authors have the option to publish the peer review history of their article (what does this mean?). If published, this will include your full peer review and any attached files.

Reviewer #1: No

Reviewer #2: No

---

## [Author Response · Author response to Decision Letter 0]

13 May 2021

Review Comments to the Author

Reviewer #1: The authors report the detection of a large deletion in the BTK gene in a large family with X-linked agammaglobulinemia. More than 500 mutations in this gene have previously been identified to cause the disease, so the paper adds one more mutation to the list. The paper is clearly written and the data analysis is well described. The authors have attempted to show that WGS is needed for identifying the pathogenic variant, however, the data presented does not support this claim (see below).

My main criticism is that the authors performed whole-exome sequencing on four affected individuals but say that they did not find "any causal variant using whole-exome sequencing". However, this is not well justified since the authors only looked at small variants identified from exome data. Given the strong association of mutations in BTK with the phenotype, the authors could easily have looked for copy number variants specifically in the BTK gene (the deletion would easily be seen as zero read depth on the X chromosome in males). There are many tools for detection of CNVs from exome data that could also have been used. Simply visualizing read depth using the IGV viewer can reveal deletions that overlap coding regions. Therefore, the claim that WGS was needed to detect the deletion in the BTK gene is not supported. WGS is definitely beneficial for identifying the precise breakpoints but exome sequencing is still 3-4 times cheaper than WGS. I would recommend that the authors report the analysis of read depth from exome data.

Response: We agree with the reviewer the point regarding visualizing the whole exome sequencing (WES) reads on the IGV could potentially identify the deletion. However in the case of WES, one cannot be sure whether such a loss of coverage could be due to chromosomal deletion or the efficiency of probes. Additionally the tools used for calling CNV using WES dataset are not accurate and robust and lead to false positive calls even in normal cases. In this context, WGS is beneficial to identify the precise breakpoint in the deletion as well as have sensitive and specific tools for structural variant calling - given it is independent of the probes, and additionally the paired-end reads can accurately identify not just the deletion but also the breakpoints. In addition, WGS provides us with an appropriate basis to perform further analysis like haplotypes and ancestry assessments which are not possible using WES dataset.

The rationale for performing chromosome painting analysis of the region around the deletion is not clear to me. The large deletion identified in this family cannot be expected to be seen in controls since it would cause a severe disease in males. Therefore, it cannot be expected to have a specific population origin. The individuals studied in this paper are from India and it is not surprising that they have South Asian ancestry. This should be true for their entire genome, rather than just the region around the deletion.

Reviewer: We apologize to the reviewer for not making it clear in the manuscript. The purpose of performing the haplotype ancestry for the patients in our study affected with PID. It is previously described in literature that a number of variants in PID have a founder effect and could be mapped to specific ancestry (PMID: 20414676, 27896283, 32822427) . The basic purpose of performing the haplotype ancestry is to identify whether the variant in the South Indian family might have a specific ancestry and also verify whether this could be associated with a founder effect in the Indian population. We have included the statement in the revised manuscript to make it clear for the reader also.

Reviewer #2: A. Is the manuscript technically sound, and the data support the conclusions?

The manuscript by Jain et al. describes about the identification of large deletion (5,296 bp) encompassing exons 3-5 of the BTK gene of the patients with X - linked agammaglobulinemia (XLA) using the whole-genome sequencing (WGS). The authors have performed whole-exome sequencing (WES) and WGS of the four and two patients with XLA, respectively. Also, the authors have performed several assays including multiplex-ligation dependent probe amplification (MLPA) assay. The manuscript can be accepted after the following minor concerns have been addressed.

In abstract section, the authors must rephrase the following sentence as they concluded (in the results section) that they could not find any variant which could correlate with the clinical features: “Whole genome sequencing led to identification of the accurate genetic mutation which could help in early diagnosis leading to improved outcomes, prevention of permanent organ damage and improved quality of life, as well as enabling prenatal diagnosis”.

Response: We thank the reviewer for the comment, we have included the statement in the Abstract section that “We could not identify any variant from the WES dataset that correlates with the clinical feature of the patient”. Also we have rephrased the concluding statement of the Abstract section. 

In the methodology section the authors can use multiple algorithms for the identification of alterations including copy number alterations using WES and WGS data of the patients with XLA.

Response: We completely agree with the reviewer on using multiple tools for identification of CNV using WES and WGS. However the tools for CNV calling in the WES dataset are not robust and results in the false negative CNV calling. While using the WGS dataset, there are multiple tools with high sensitivity and specificity for CNV calling. In our study, we have used a tool LUMPY (version 0.2.13) for structural variant calling that identify the large deletion in our patient

B. Has the statistical analysis been performed appropriately and rigorously?

The statistical analysis has not been performed by the authors.

Response: Since the study involved individual cases , no statistical analysis was performed 

C. Have the authors made all data underlying the findings in their manuscript fully available?

The authors have provided analyzed data in the figures/tables and supplementary information in the manuscript wherever necessary. The raw data can be accessed through the convener of the Institutional Human Ethics Committee of the institute.

Response: Since the WGS/WES constitute data that can be potentially identifiable and contains patient sensitive information, the data is available only on request. Access to the data could be requested by mailing to Dr. Jyoti Yadav (ni.ser.bigi@vaday.j) who is the convener of the Institutional Human Ethics Committee of CSIR-IGIB.

D. Is the manuscript presented in an intelligible fashion and written in standard English?

Overall manuscript is okay written, however the authors can rephrase the sentences in the methodology section for example, “small indels, small insertion, and small deletion”.

Response: We thank the reviewer we have rephrased by removing the small indels in the statement.

---

## [Decision Letter · Decision Letter 1]

28 Jun 2021

Whole genome sequencing identifies novel structural variant in a large Indian family affected with X-linked agammaglobulinemia.

PONE-D-20-38741R1

Dear Dr. Scaria,

We’re pleased to inform you that your manuscript has been judged scientifically suitable for publication and will be formally accepted for publication once it meets all outstanding technical requirements.

Kind regards,

Obul Reddy Bandapalli, MSc, PhD

Academic Editor

PLOS ONE

Additional Editor Comments (optional):

Reviewers' comments:

Reviewer's Responses to Questions

**Comments to the Author**

1. If the authors have adequately addressed your comments raised in a previous round of review and you feel that this manuscript is now acceptable for publication, you may indicate that here to bypass the “Comments to the Author” section, enter your conflict of interest statement in the “Confidential to Editor” section, and submit your "Accept" recommendation.

Reviewer #1: All comments have been addressed

Reviewer #2: All comments have been addressed

2. Is the manuscript technically sound, and do the data support the conclusions?

Reviewer #1: (No Response)

Reviewer #2: Yes

3. Has the statistical analysis been performed appropriately and rigorously? 

Reviewer #1: (No Response)

Reviewer #2: N/A

4. Have the authors made all data underlying the findings in their manuscript fully available?

Reviewer #1: (No Response)

Reviewer #2: Yes

5. Is the manuscript presented in an intelligible fashion and written in standard English?

Reviewer #1: (No Response)

Reviewer #2: Yes

6. Review Comments to the Author

Reviewer #1: (No Response)

Reviewer #2: (No Response)

7. PLOS authors have the option to publish the peer review history of their article (what does this mean?). If published, this will include your full peer review and any attached files.

Reviewer #1: No

Reviewer #2: No

---

## [Editor Report · Acceptance letter]

1 Jul 2021

PONE-D-20-38741R1 

Whole genome sequencing identifies novel structural variant in a large Indian family affected with X-linked agammaglobulinemia 

Dear Dr. Scaria:

I'm pleased to inform you that your manuscript has been deemed suitable for publication in PLOS ONE. Congratulations! Your manuscript is now with our production department. 

Kind regards, 

on behalf of

Dr. Obul Reddy Bandapalli 

Academic Editor

PLOS ONE